# Unlabeled Data vs. Pre-trained Knowledge: Rethinking Semi-Supervised Learning in the Era of Large Models

## Abstract

Semi-supervised learning (SSL) alleviates the cost of data labeling process by exploiting unlabeled data and has achieved promising results. Meanwhile, with the development of large foundation models, exploiting pre-trained models becomes a promising way to address the label scarcity in the downstream tasks, such as various parameter-efficient fine-tuning techniques. This raises a natural yet critical question: When labeled data is limited, should we rely on unlabeled data or pre-trained models? To investigate this issue, we conduct a fair comparison between SSL methods and pre-trained models (e.g., CLIP) on representative image classification tasks under a controlled supervision budget. Experiments reveal that SSL has met its "Waterloo" in the era of large models, as pre-trained models show both high efficiency and strong performance on widely adopted SSL benchmarks. This underscores the urgent need for SSL researchers to explore new avenues, such as deeper integration between the SSL and pre-trained models. Furthermore, we investigate the potential of Multi-Modal Large Language Models (MLLMs) in image classification tasks. Results show that, despite their massive parameter scales, MLLMs still face significant performance limitations, highlighting that even a seemingly well-studied task remains highly challenging.

## 1 Introduction

Deep neural networks have demonstrated performance comparable to that of humans in certain supervised learning tasks, such as image classification (Geirhos et al., 2018; He et al., 2016; Lee et al., 2013b). However, these impressive results rely heavily on large amounts of labeled training data. In many real-world applications, obtaining such labeled data often demands substantial human and financial resources (Yang et al., 2022). To reduce the dependence on labeled data and thereby cut costs, researchers have proposed **Semi-Supervised Learning (SSL)**, leveraging unlabeled data to enhance model performance for image classification tasks.

In recent years, **Pre-trained Models** have attracted considerable attention. Among them, vision-language models (VLMs) such as CLIP (Radford et al., 2021), which achieves semantic alignment during pre-training, has become a widely adopted backbone for image classification tasks. Building upon CLIP, various fine-tuning strategies (e.g., prompt tuning (Zhou et al., 2022b)) have been proposed to enhance task-specific performance using minimal labeled data by leveraging the rich knowledge acquired during pre-training (Zhou et al., 2022b; Khattak et al., 2023a;b). More recently, multi-modal large language models (MLLMs) such as Qwen-VL (Bai et al., 2023) and GPT-4o-min (Hurst et al., 2024) have emerged, featuring significantly larger parameter scales and stronger general capabilities in visual tasks.

As shown in Figure 1, both SSL and pre-trained models are well suited to image classification tasks with limited labeled data, by leveraging unlabeled data and pre-trained knowledge, respectively. This makes us come up with three questions naturally:

1. When labels are scarce, should we prioritize utilizing unlabeled data or pre-trained models?

2. What are the strengths and limitations of leveraging unlabeled data versus exploiting pre-trained models?

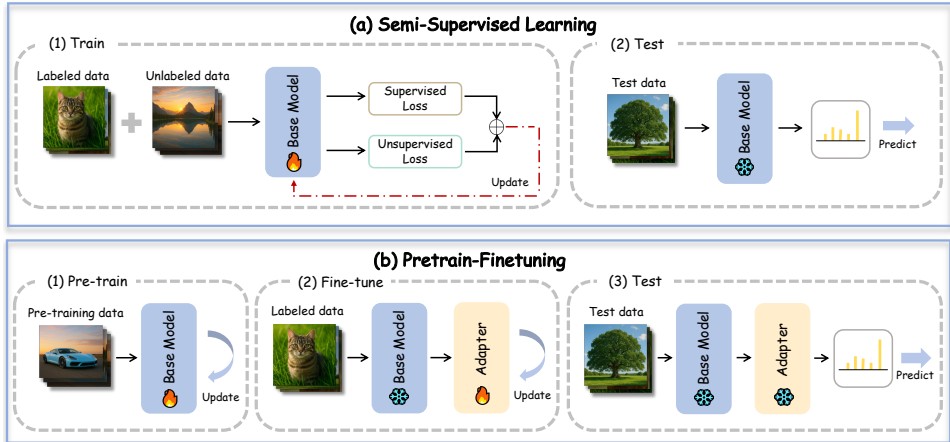

Figure 1: Pipelines of semi-supervised learning and pretrain-finetuning.

3. How can we design methods that mitigate the shortcomings of both approaches while maximizing their complementary benefits?

To answer these questions, we adopt widely used CLIP-based VLMs as representative pre-trained models, owing to their strong generalization capabilities and proven effectiveness in zero- and few-shot learning across diverse domains (Radford et al., 2021; Zhou et al., 2022b; Khattak et al., 2023b). We extracted and designed a fair evaluation framework based on papers from top-tier conferences and journals to compare various SSL methods and pre-trained models under consistent experimental conditions. Specifically, we standardize the amount of labeled data used across different experimental setups, including semi-supervised, open-set semi-supervised, open-world semi-supervised, and long-tailed semi-supervised settings, ensuring equal supervision signals for all approaches. Furthermore, to further enrich our experiments, we also included a comparison of performance on a less common medical dataset.

The experimental results show that the SSL method faces its Waterloo, with problems such as low training efficiency and poor performance. Moreover, when the test data contains unseen classes, SSL struggles to learn and adapt effectively. In contrast, pre-trained models, endowed with rich prior knowledge, can quickly adapt to downstream tasks with limited data, demonstrating strong generalization to open-world scenarios and long-tailed distributions. Although pre-trained models may suffer from adaptation bias due to distribution discrepancies between pre-training data and downstream tasks, these issues can be alleviated through additional training or data distillation techniques. Such approaches also yield performance gains and higher efficiency compared to SSL methods.

To bridge the gap between the two paradigms, we further explore hybrid algorithms. By leveraging their generalization ability, these methods can generate more reliable pseudo-labels, thereby substantially enhancing the efficiency and stability of semi-supervised training. We explore the performance of larger-scale MLLMs on image classification tasks, and the results show that they are likewise constrained by adaptation bias and other issues arising from the limitations of their visual encoders. To further advance research in this area, we propose the following directions:

1. It is no longer sufficient for SSL researchers to compare only with previous SSL methods in image classification; systematic comparisons with pre-trained models are necessary to ensure fair and meaningful evaluations.

2. Exploring effective strategies to integrate SSL techniques with the adaptation of pre-trained models represents another promising approach to enhance both the efficiency of SSL and the adaptability of pre-trained models.

3. Exploring the capabilities of MLLMs on image classification tasks, particularly across varying distributions and resolutions, as experiments highlight that even a seemingly well-studied task like image classification remains challenging for MLLMs.

## 2  RELATED WORK

**Semi-Supervised Learning.** The goal of Semi-Supervised Learning is to reduce the cost of data annotation by leveraging unlabeled data, under the assumption that both labeled and unlabeled samples belong to a predefined set of known classes. The primary objective is to assign accurate ground-truth labels to unlabeled instances. Existing SSL methods can be broadly categorized into three main approaches: Hybrid and Holistic Methods (Berthelot et al., 2019a;b). Pseudo-Labeling assigns pseudo-labels to unlabeled data based on the model's own predictions throughout training (Lee et al., 2013a). FixMatch (Sohn et al., 2020) combines regularization and confidence-based filtering by enforcing consistency between weakly and strongly augmented samples, while only using high-confidence pseudo-labels for training, resulting in significant performance gains with limited labeled data.

**Pre-trained Models.** Recent advances in pre-trained models have attracted considerable attention, particularly with the emergence of VLMs such as CLIP (Radford et al., 2021) and MLLMs such as Qwen-VL (Bai et al., 2023). While prior work (Zoph et al., 2020) has compared pre-trained models with self-training algorithms in object detection and segmentation tasks, it did not consider VLMs, MLLMs, or even SSL settings. To address this gap, we focus on the widely studied image classification task and systematically investigate the performance of VLMs and their fine-tuning strategies. As the pioneering work for fine-tuning VLMs, CoOp (Zhou et al., 2022b) for the first time introduces learnable prompts to transfer the task-specific knowledge to VLMs. Building on this idea, MaPLe (Khattak et al., 2023a) and PromptSRC (Khattak et al., 2023b) extend prompt tuning to both visual and textual encoders, enabling joint vision-language adaptation. Multi-modal large language models, on the other hand, are primarily geared toward reasoning tasks, and their capabilities in image classification remain underexplored. Therefore, this paper selects three MLLMs for a comparative study.

## 3  EVALUATION AND ARCHITECTURE SETTINGS

We summarize four common SSL settings from 50 recent top-tier publications, which represent the primary and most challenging focuses of current research. Then, we reformulate these settings for pre-trained VLMs to ensure fair comparisons. The dataset $\mathcal{D}$ used in this setting consists of a small labeled subset $\mathcal{D}_l = \{(\mathbf{x}_i, y_i)\}_{i=1}^{N_l}$ and a larger unlabeled subset $\mathcal{D}_u = \{\mathbf{x}_j\}_{j=1}^{N_u}$, where the number of labeled samples is significantly smaller than the unlabeled samples ($N_l \ll N_u$). The following provides an overview of each setting, and details are shown in Appendix B.

- **Standard Semi-Supervised Learning (Sohn et al., 2020).** The core feature of this setting is that **the labeled and unlabeled datasets share identical classes, both belonging to the same known class set** $\mathcal{Y}^l$. Its primary goal is to leverage unlabeled data to assist a small amount of labeled data in improving model generalization performance. **Since pre-trained models do not utilize unlabeled data, our open-set setting reduces to the semi-supervised setting.**

- **Open-Set Semi-Supervised Learning (Oliver et al., 2018).** In this setting, the unlabeled dataset $\mathcal{D}_u$ may include instances from unseen classes, i.e., $\mathcal{Y}^l \subset \mathcal{Y}^u$ with $\mathcal{Y}^{new} = \mathcal{Y}^u \setminus \mathcal{Y}^l$. The core challenge is to **enhance the classification performance of known classes while enabling the model to distinguish between known and unseen classes**. Since pre-trained models do not involve any unlabeled data during training, they do not need to handle interference from unseen classes in unlabeled data. Thus, **this open-set semi-supervised setting is essentially equivalent to the standard semi-supervised learning scenario** for pre-trained models, which can be adapted merely through fine-tuning on a small amount of labeled data without the need for additional mechanisms designed for unseen class detection or separation.

- **Open-World Semi-Supervised Learning (Niu et al., 2024).** This setting extends the open-set scenario by aiming to classify both known and novel classes. with the core goal of **not only distinguishing between known and unseen classes but also effectively classifying unseen (novel) classes**. It requires models to possess the dual capability of "recognizing known classes + discovering and classifying novel classes." Pre-trained models inherently have the advantage of adapting to this scenario—through cross-modal alignment capabilities acquired from pre-training on large-scale image-text pairs, they can support zero-shot classification. Without exposure to novel class samples during training, pre-trained models can generalize to novel

Table 1: Accuracy on CIFAR-10, CIFAR-100, and STL-10. The best performance for each dataset is in bold, while the top scores in the semi-supervised setting are underlined.

| Method | CIFAR-10 | | | CIFAR-100 | | | STL-10 |
|---|---|---|---|---|---|---|---|
| | 40 labels | 250 labels | 4000 labels | 400 labels | 2500 labels | 10000 labels | 1000 labels |
| MixMatch | $52.46_{\pm 11.50}$ | $88.95_{\pm 0.86}$ | $93.58_{\pm 0.10}$ | $32.39_{\pm 1.32}$ | $60.06_{\pm 0.37}$ | $71.69_{\pm 0.33}$ | $89.59_{\pm 0.61}$ |
| UDA | $70.95_{\pm 5.93}$ | $91.18_{\pm 1.08}$ | $95.12_{\pm 0.18}$ | $40.72_{\pm 0.88}$ | $66.87_{\pm 0.22}$ | $75.50_{\pm 0.25}$ | $92.34_{\pm 0.56}$ |
| ReMixMatch | $80.90_{\pm 9.64}$ | $94.56_{\pm 0.05}$ | $95.28_{\pm 0.13}$ | $55.72_{\pm 2.06}$ | $\underline{\mathbf{72.57}}_{\pm 0.31}$ | $76.97_{\pm 0.56}$ | $94.77_{\pm 0.45}$ |
| FixMatch (RA) | $86.19_{\pm 3.37}$ | $94.93_{\pm 0.65}$ | $\underline{\mathbf{95.74}}_{\pm 0.05}$ | $51.15_{\pm 1.75}$ | $71.71_{\pm 0.11}$ | $\underline{\mathbf{77.40}}_{\pm 0.12}$ | $92.02_{\pm 1.50}$ |
| FixMatch (CTA) | $\underline{\mathbf{88.61}}_{\pm 3.35}$ | $\underline{94.93}_{\pm 0.33}$ | $95.69_{\pm 0.15}$ | $50.05_{\pm 3.01}$ | $71.36_{\pm 0.24}$ | $76.82_{\pm 0.11}$ | $\underline{94.83}_{\pm 0.63}$ |
| ZSCLIP | | $79.30_{\pm 0.00}$ | | | $46.00_{\pm 0.00}$ | | $97.40_{\pm 0.00}$ |
| CoOp | $80.60_{\pm 1.79}$ | $85.10_{\pm 0.29}$ | $89.13_{\pm 0.12}$ | $49.00_{\pm 0.83}$ | $57.43_{\pm 0.25}$ | $61.40_{\pm 0.14}$ | $98.23_{\pm 0.17}$ |
| PromptSRC | $86.03_{\pm 0.48}$ | $90.10_{\pm 0.08}$ | $93.90_{\pm 0.14}$ | $\mathbf{60.33}_{\pm 0.17}$ | $66.33_{\pm 0.29}$ | $70.10_{\pm 0.22}$ | $\mathbf{98.57}_{\pm 0.12}$ |

classes solely through textual descriptions of class names. Therefore, **this open-world semi-supervised setting is highly similar to the widely used "base-to-novel" setting in the VLM field (Zhou et al., 2022b;a; Khattak et al., 2023a)**.

- **Long-Tailed Semi-Supervised Learning (Kukleva et al., 2023).** This setting features a highly imbalanced class distribution, where major (head) classes have many samples and minor (tail) classes have few. The **imbalance ratio (IR)** is defined as the sample count ratio between the largest and smallest classes. Since VLMs use a fixed number of samples per class in few-shot settings, aligning **the shot count with tail classes** reduces the problem to a standard few-shot scenario, enabling fair comparison.

**Evaluation Setting of Comparisons between VLMs and MLLMs.** To align with the characteristics of the two model types, a differentiated adaptation scheme is adopted in the experiment: for VLMs, we use the widely accepted Few-shot learning setup in the field to evaluate their few-shot adaptation capability; for MLLMs, we explicitly inform them of the optional class set for the task via prompts, guiding the models to output classification results directly. Details regarding the prompt design and output format specifications for MLLMs can be found in Appendix D.

**Architecture Settings.** Different methods typically correspond to more suitable model architectures, a key prerequisite for ensuring experimental validity. Thus, we adhere to the well-recognized standard configuration conventions in the field: the RN28 architecture is adopted for SSL methods when trained on low-resolution datasets like CIFAR, while the RN50 architecture is selected for ImageNet. All experiments are reproduced in accordance with the optimal experimental conditions for each task. For CLIP-based methods, since our work is the first to simultaneously compare multiple semi-supervised settings, we use the ViT-B/16 architecture by default. For Multimodal Large Language Models (MLLMs), we conduct tests via their official APIs.

## 4 EXPERIMENTS

### 4.1 BASELINES SELECTION

Given that SSL is primarily applied to image classification tasks, we selected FixMatch (Sohn et al., 2020) as a representative method for standard semi-supervised learning, DWD (Ban et al., 2024) for open-set semi-supervised learning, OwMatch (Niu et al., 2024) for open-world semi-supervised learning, and CCL (Zhou et al., 2024) for long-tailed semi-supervised learning. For pre-trained models, we consider CLIP (Radford et al., 2021) along with its prompt tuning variants: CoOp, which was the first to introduce prompt tuning for CLIP (Zhou et al., 2022b), and PromptSRC (Khattak et al., 2023b), a state-of-the-art method in this line of work.

### 4.2 COMPARISONS IN STANDARD SSL SETTING

**Datasets and Experimental Setup.** We evaluate both SSL and pre-trained VLMs on CIFAR-10/100 (Krizhevsky et al., 2009), STL-10 (Coates et al., 2011), and ImageNet (Deng et al., 2009),

Table 2: Performance of SSL and pre-trained VLMs on the ImageNet dataset.

| Method | UDA | FixMatch | ZSCLIP | CoOp | PromptSRC |
|---|---|---|---|---|---|
| Label | 10% labels | 10% labels | 0 labels | 2% labels | 2% labels |
| Accuracy | $68.78_{\pm0.43}$ | $71.46_{\pm0.52}$ | $66.70_{\pm0.00}$ | $73.43_{\pm0.12}$ | $\mathbf{74.03}_{\pm0.09}$ |

Table 3: Performance of SSL and pre-trained VLMs on the ISIC2018 dataset.

| Method | Accuracy | Sensitivity (TPR) | Specificity (TNR) | F1 |
|---|---|---|---|---|
| Self-training | 95.02 | 79.15 | 93.86 | 73.05 |
| FlexMatch (RA) | 93.40 ± 0.05 | 71.25 ± 0.23 | 92.23 ± 0.17 | 61.37 ± 0.22 |
| CoMatch (RA) | 93.39 ± 0.04 | 70.22 ± 0.25 | 92.29 ± 0.11 | 61.94 ± 0.21 |
| FixMatch (RA) | 93.23 ± 0.14 | 72.91 ± 0.52 | 92.21 ± 0.03 | 60.19 ± 0.31 |
| Zero-shot CLIP | 14.6 | 22.4 | 86.4 | 11.1 |
| CoOp | 55.43±2.49 | 52.63±2.32 | 92.33±0.50 | 41.80±2.01 |
| PromptSRC | 59.97±2.09 | 60.70±2.87 | 93.17±0.37 | 51.10±1.77 |

under varying annotation levels and augmentation strategies. In particular, we selected the ISIC2018 medical image dataset (Codella et al., 2018), which is more scarce in real-world scenarios, for testing, aiming to achieve a more comprehensive comparison (Zhou et al., 2023b). For VLMs, we simulate few-shot learning by training on the same labeled data only, without accessing unlabeled samples. Full implementation details are provided in Appendix B.2.

**Experiment Results.** As shown in Table 1, SSL methods generally perform well on low-resolution datasets such as CIFAR-10 and CIFAR-100, which have an image resolution of $32 \times 32$, surpassing zero-shot CLIP and its fine-tuning variants. This advantage stems from the challenge of extracting meaningful features from low-resolution inputs — a core reason being the "adaptation bias" (induced by resolution discrepancies) between the pre-trained data of VLMs and the target downstream tasks. In contrast, pre-trained VLMs perform exceptionally well on higher-resolution datasets such as STL-10 ($96 \times 96$): even the zero-shot CLIP model outperforms the best-performing SSL baselines. In table 3, on the new dataset we introduced, CLIP-based methods suffer from visual biases caused by differences in image distribution. Even after fine-tuning, they remain far less effective than SSL methods. These results further support our core argument: the impressive generalization capabilities of pre-trained models are inherently *conditional*, dependent on the consistency between the pre-training distribution and the downstream distribution. When this consistency is weak, SSL remains a powerful and sometimes superior alternative.

Moreover, as shown in Table 2, on the ImageNet dataset, FixMatch requires a significantly larger proportion of labeled data (around 10%) but still lags behind in accuracy. By comparison, VLMs achieve over 73% accuracy using only 2% of labeled samples, with performance improving as more labeled data becomes available. In addition to accuracy, training SSL models on CIFAR datasets takes GPU hours (Berthelot et al., 2019a; Sohn et al., 2020), whereas prompt tuning approaches for VLMs (Khattak et al., 2023a;b; Zhou et al., 2022b) can be completed in just a few GPU minutes. In summary, the above observations highlight the complementary strengths of SSL and pre-trained models, with the latter generally offering a higher performance ceiling and faster training speed.

**Analysis of "Adaptation Bias".** We argue that "adaptation bias" is typically caused by two main factors: the capacity of the visual encoder and discrepancies between training and testing styles (e.g., differences in resolution and image style). To evaluate the impact of resolution, we test CLIP models with different backbones on the CIFAR-10/100 datasets. As shown in Table 2, performance improves steadily with the number of visual encoder parameters, indicating that the suboptimal results on low-resolution images are not due to the encoder itself. Since the original training data of CLIP is unavailable, we approximate this bias through input resolution (CLIP is fixed at an input resolution of $224 \times 224$). We then evaluated various methods under different input resolutions on the STL-10 dataset (Figure 3). As the resolution increases, model performance consistently improves, and adaptation bias is gradually reduced. This suggests that the distribution shift between training and testing is the fundamental cause of "adaptation bias". Fine-tuning can partially mitigate the

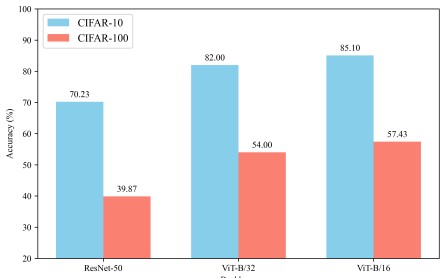 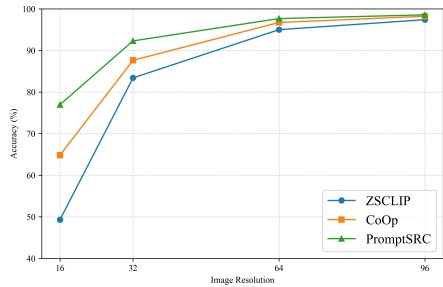

Figure 2: Performance of different backbones.  Figure 3: Performance at different resolutions.

Table 4: Performance comparison over three datasets. We report the mean accuracy averaged over three seeds, along with standard error.

| Dataset | FixMatch | MPL | IOMatch | DWD-SL | ZSCLIP | CoOp | PromptSRC |
|---|---|---|---|---|---|---|---|
| CIFAR-10/100 | $78.91_{\pm0.15}$ | $70.95_{\pm0.34}$ | $77.66_{\pm0.22}$ | $80.05_{\pm0.14}$ | $79.30_{\pm0.00}$ | $87.87_{\pm0.26}$ | $\mathbf{92.27}_{\pm0.12}$ |
| ImageNet-30 | $70.07_{\pm0.26}$ | $72.65_{\pm0.70}$ | $79.23_{\pm0.29}$ | $\underline{82.20}_{\pm0.38}$ | $98.80_{\pm0.00}$ | $99.33_{\pm0.12}$ | $\mathbf{99.30}_{\pm0.00}$ |
| ImageNet-100 | $65.11_{\pm0.32}$ | $68.43_{\pm0.33}$ | $66.85_{\pm0.19}$ | $\underline{82.81}_{\pm0.31}$ | $87.70_{\pm0.00}$ | $91.37_{\pm0.12}$ | $\mathbf{91.70}_{\pm0.08}$ |

adaptation bias of models toward low-resolution images (e.g., PromptSRC outperforms CLIP at $16 \times 16$ resolution). Additionally, model distillation techniques have been shown to alleviate this bias (Lee et al., 2025; Zhou et al., 2023a). For instance, RADIO (Heinrich et al., 2024) enhances the stability and performance of student models across varying input resolutions by distilling knowledge from teacher models and incorporating strategies such as multi-resolution training and balanced teacher losses.

> **Takeaway 1:** *Pre-trained models demonstrate strong generalization capabilities and high data efficiency, enabling them to outperform traditional SSL methods with only a small amount of labeled data.* *Due to the presence of "adaptation bias" in pre-trained models, solving practical problems requires selecting a method based on the magnitude of the distribution difference.*

### 4.3 COMPARISONS IN OPEN-SET SSL SETTING

**Datasets and Experimental Setup.** We evaluate both paradigms on standard open-set benchmarks, including CIFAR-10/100 (Krizhevsky et al., 2009), ImageNet-30 (Hendrycks et al., 2019), and ImageNet-100 (Cao et al., 2021). For CIFAR-10/100, we use 100 labeled images per class from CIFAR-10 as the supervised data and the entire CIFAR-100 dataset as the unsupervised data. For ImageNet-30 and ImageNet-100, we adopt a 50-shot setting, using 50 labeled samples per class from the selected in-distribution (ID) classes. It is important to note that the test set in this setting contains only the classes seen during training. For pre-trained models, the experimental setups are consistent with those in the semi-supervised setting. Further details can be found in Appendix B.3.

**Experiment Results.** As shown in Table 4, zero-shot CLIP and its fine-tuned variants consistently outperform other methods across all datasets, with particularly strong results on ImageNet-30 and ImageNet-100. This advantage stems from their data efficiency—without reliance on noisy unlabeled data, VLMs are less impacted by out-of-distribution (OOD) samples in low-quality datasets. In contrast, SSL methods are more sensitive to such data quality issues, as they require specialized mechanisms to identify and filter out OOD samples. This reliance on additional OOD handling procedures inevitably affects both training efficiency and final performance, often leading to suboptimal results. Moreover, pre-trained models derive distinct advantages from two key stages: first, they undergo extensive large-scale pre-training on diverse datasets, and second, they are refined through targeted fine-tuning on task-specific data. This two-step process enables them to achieve consistently lower performance variance across different data distributions and a higher performance ceiling.

Table 5: Average accuracy on CIFAR-10/100 and ImageNet-100 datasets with both new class ratio and label ratio of 50%.

| Method | CIFAR-10 | | | CIFAR-100 | | | ImageNet-100 | | |
|---|---|---|---|---|---|---|---|---|---|
| | Base | New | All | Base | New | All | Base | New | All |
| FixMatch | 71.5 | 50.4 | 49.5 | 39.6 | 23.5 | 20.3 | 65.8 | 36.7 | 34.9 |
| DS³L | 77.6 | 45.3 | 40.2 | 55.1 | 23.7 | 24.0 | 71.2 | 32.5 | 30.8 |
| CGDL | 72.3 | 44.6 | 39.7 | 49.3 | 22.5 | 23.6 | 67.3 | 33.8 | 31.9 |
| DTC | 53.9 | 39.5 | 38.3 | 31.3 | 22.9 | 18.3 | 25.6 | 20.8 | 21.3 |
| RankStats | 86.6 | 81.0 | 82.9 | 36.4 | 28.4 | 23.1 | 47.3 | 28.7 | 40.3 |
| SimCLR | 58.3 | 63.4 | 51.7 | 28.6 | 21.1 | 22.3 | 39.5 | 35.7 | 36.9 |
| ORCA | 88.2 | 90.4 | 89.7 | 66.9 | 43.0 | 48.1 | 89.1 | 72.1 | 77.8 |
| NACH | 89.5 | 92.2 | 91.3 | 68.7 | 47.0 | 52.1 | 91.0 | 75.5 | 79.6 |
| OpenLDN | 95.7 | 95.1 | 95.4 | 73.5 | 46.8 | 60.1 | 89.6 | 68.6 | 79.1 |
| OpenCon | 89.3 | 91.1 | 90.4 | 69.1 | 47.8 | 52.7 | 90.6 | 80.8 | 83.8 |
| OwMatch | 93.0 | 95.9 | 94.4 | 74.5 | 55.9 | 65.1 | **91.7** | 72.0 | 81.8 |
| OwMatch+ | **96.5** | **97.1** | **96.8** | **80.1** | **63.9** | **71.9** | 91.5 | 79.6 | 85.5 |
| ZSCLIP | 88.0 | 88.1 | 79.3 | 57.4 | 53.1 | 46.0 | 89.5 | 87.9 | 87.7 |
| CoOp | 91.1 | 90.4 | 81.3 | 65.4 | 51.0 | 48.4 | 89.7 | 83.2 | 85.5 |
| PromptSRC | 92.9 | 90.0 | 83.8 | 72.3 | 62.8 | 58.9 | 91.2 | **89.7** | **89.4** |

**Takeaway 2:** *When the unlabeled data contains out-of-distribution classes, pre-trained models can be leveraged to avoid unlabeled training, capitalizing on their pre-trained knowledge and rapid adaptation capabilities. Additionally, pre-trained models offer advantages such as robust result stability and high performance.*

## 4.4 COMPARISONS IN OPEN-WORLD SSL SETTING

**Datasets and Experimental Setup.** We conduct evaluations on CIFAR-10/100 (Krizhevsky et al., 2009), and ImageNet-100 (Cao et al., 2021). In each dataset, the classes are divided into known (base) and new classes, with the proportion of new classes set to 50% by default. A portion of the samples from the known classes are labeled according to a given label ratio, while the remaining part, together with all the new class data, formed the unlabeled set. This setup requires the model to distinguish and classify both base and new classes simultaneously. For more detailed settings, please refer to Appendix B.4.

**Experiment Results.** In the open-world setting, SSL methods aim to identify unseen classes from unlabeled data during training, typically using clustering-based pseudo-labeling, while evaluating both base and novel classes at test time. As shown in Table 5, SSL methods achieve the best performance on CIFAR-10 and CIFAR-100 datasets, indicating that specially tailored losses outperform pretrained models affected by "adaptation bias", although requiring substantial GPU training time. In contrast, on the ImageNet-100 dataset, zero-shot CLIP already approaches the performance of OwMatch on base classes and significantly outperforms it on novel classes. This can be attributed to the richer feature representations captured by pre-trained models as image resolution increases, thereby reducing "adaptation bias". Notably, fine-tuned models exhibit a smaller performance gap between base and novel classes on ImageNet-100 compared to SSL methods, highlighting their strong generalization capabilities. In summary, pre-trained models that overcome adaptation bias offer advantages such as rapid adaptability and strong generalization, whereas SSL methods require careful design to effectively handle both novel and unseen classes.

**Takeaway 3:** *In the open-world setting, SSL methods perform better on low-resolution datasets, while pre-trained VLMs demonstrate balanced performance on both base and new classes in high-resolution datasets. Combined with our open-set analysis, we conclude that the pre-trained models are better suited for scenarios where unlabeled data may contain unseen classes.*

Table 6: Performance of SSL and VLMs under consistent imbalance ratio setting.

| Method | CIFAR10-LT | | | | CIFAR100-LT | | | |
|---|---|---|---|---|---|---|---|---|
| | $\gamma = \gamma_l = \gamma_u = 100$ | | $\gamma = \gamma_l = \gamma_u 150$ | | $\gamma = \gamma_l = \gamma_u = 10$ | | $\gamma = \gamma_l = \gamma_u = 20$ | |
| | $N_1 = 500$ | $N_1 = 1500$ | $N_1 = 500$ | $N_1 = 1500$ | $N_1 = 50$ | $N_1 = 150$ | $N_1 = 50$ | $N_1 = 150$ |
| | $M_1 = 4000$ | $M_1 = 3000$ | $M_1 = 4000$ | $M_1 = 3000$ | $M_1 = 400$ | $M_1 = 300$ | $M_1 = 400$ | $M_1 = 300$ |
| Supervised | $47.3_{\pm0.95}$ | $61.9_{\pm0.41}$ | $44.2_{\pm0.33}$ | $58.2_{\pm0.29}$ | $29.6_{\pm0.57}$ | $46.9_{\pm0.22}$ | $25.1_{\pm1.14}$ | $41.2_{\pm0.15}$ |
| w/ LA | $53.3_{\pm0.44}$ | $70.6_{\pm0.21}$ | $49.5_{\pm0.40}$ | $67.1_{\pm0.78}$ | $30.2_{\pm0.44}$ | $48.7_{\pm0.89}$ | $26.5_{\pm1.31}$ | $44.1_{\pm0.42}$ |
| FixMatch + LA | $75.3_{\pm2.45}$ | $82.0_{\pm0.36}$ | $67.0_{\pm2.49}$ | $78.0_{\pm0.91}$ | $47.3_{\pm0.58}$ | $58.6_{\pm0.36}$ | $41.4_{\pm0.93}$ | $53.4_{\pm0.32}$ |
| w/ DARP | $76.6_{\pm0.92}$ | $80.8_{\pm0.62}$ | $68.2_{\pm0.24}$ | $76.7_{\pm1.35}$ | $50.4_{\pm0.65}$ | $59.0_{\pm0.53}$ | $43.6_{\pm1.05}$ | $55.3_{\pm0.84}$ |
| w/ DASO | $77._{\pm0.8}8$ | $82.5_{\pm0.60}$ | $71.0_{\pm1.68}$ | $79.0_{\pm2.23}$ | $50.1_{\pm0.75}$ | $61.0_{\pm0.72}$ | $43.5_{\pm0.81}$ | $56.1_{\pm0.72}$ |
| FixMatch + ACR | $81.6_{\pm0.19}$ | $84.1_{\pm0.20}$ | $77.0_{\pm1.19}$ | $82.0_{\pm0.41}$ | $51.1_{\pm0.80}$ | $61.3_{\pm0.26}$ | $46.1_{\pm0.34}$ | $56.5_{\pm0.33}$ |
| FixMatch + CPE | $80.7_{\pm0.96}$ | $84.4_{\pm0.29}$ | $76.8_{\pm0.53}$ | $82.3_{\pm0.36}$ | $50.6_{\pm0.45}$ | $59.8_{\pm0.13}$ | $43.6_{\pm0.88}$ | $55.8_{\pm0.65}$ |
| FixMatch + CCL | $\underline{84.5}_{\pm0.38}$ | $\underline{86.2}_{\pm0.35}$ | $\underline{81.5}_{\pm0.99}$ | $\underline{84.0}_{\pm0.21}$ | $\underline{53.5}_{\pm0.49}$ | $\underline{63.5}_{\pm0.39}$ | $\underline{46.8}_{\pm0.45}$ | $\underline{57.5}_{\pm0.55}$ |
| ZSCLIP | $79.30_{\pm0.00}$ | | | | $46.00_{\pm0.00}$ | | | |
| CoOp | $80.73_{\pm1.90}$ | $84.20_{\pm0.50}$ | $80.60_{\pm1.79}$ | $82.23_{\pm0.75}$ | $49.83_{\pm1.04}$ | $55.23_{\pm0.50}$ | $44.83_{\pm0.45}$ | $52.57_{\pm0.12}$ |
| PromptSRC | $\mathbf{86.50}_{\pm0.37}$ | $\mathbf{88.70}_{\pm0.08}$ | $\mathbf{86.03}_{\pm0.48}$ | $\mathbf{87.67}_{\pm0.42}$ | $\mathbf{60.33}_{\pm0.17}$ | $\mathbf{64.37}_{\pm0.34}$ | $\mathbf{58.03}_{\pm0.46}$ | $\mathbf{62.83}_{\pm0.48}$ |

## 4.5 COMPARISONS IN LONG-TAILED SSL SETTING

**Datasets and Experimental Setup.** We compare different methods on four long-tailed datasets: CIFAR-10-LT (Krizhevsky et al., 2009), CIFAR-100-LT (Krizhevsky et al., 2009), STL-10-LT (Coates et al., 2011), and ImageNet-127 (Fan et al., 2022). For synthetic long-tailed datasets (CIFAR, STL-10), the class-wise labeled/unlabeled sample counts follow an exponential decay controlled by imbalance ratios $\gamma_l$ and $\gamma_u$ (Wei & Gan, 2023). For ImageNet-127, a naturally long-tailed dataset, we follow (Zhou et al., 2024) and use 10% of the training data as labeled. To ensure fair comparison, we adopt the same splits and protocols as SSL methods (Niu et al., 2024), and report performance of pre-trained models in a few-shot setting where the number of shots is determined by the smallest labeled class (e.g., 5-shot for CIFAR-10 when $\gamma_l = 100$ and $N_1 = 500$). Full setup details and shot derivations are available in Appendix B.5.

**Experiment Results.** As shown in Table 6, in the long-tailed semi-supervised setting, SSL methods are designed to ensure that the learning process effectively captures all classes, even when the distribution is dominated by a few head classes. However, as shown in Table 1, their performance often declines in the same datasets compared to standard SSL, highlighting their ongoing vulnerability to the challenges of long-tailed distributions. In contrast, the evaluation results of pre-trained models underscore their efficiency, which can be fine-tuned with only a small amount of labeled data and still perform well, even on low-resolution datasets, a common challenge for these models.

We also conducted experiments about STL-10 and Imagenet-127 datasets (detailed results are shown in Appendix C), and concluded that pre-trained models perform well on imbalanced datasets but struggle with semantically coarse tasks due to ambiguous class definitions. Although downstream fine-tuning can partially mitigate this issue, the underlying semantic constraints remain. We provide a more in-depth explanation and visualization of this phenomenon in Appendix C. Ultimately, we concluded that the highly abstract nature of downstream task categories inherently leads to "adaptation bias", a fundamental issue that negatively impacts the performance of VLMs.

> **Takeaway 4:** *The low data requirements of pre-trained models allow them to effectively address long-tailed distributions, offering a promising solution to real-world data imbalance. However, when the semantic boundaries of the downstream tasks are not clear, pre-trained VLMs' adaptability will decline.*

## 4.6 COMPARISONS BETWEEN VLMS AND MLLMS

**Datasets and Experimental Setup.** We compare the performance of CLIP with existing MLLM approaches such as Qwen and GPT-4o-mini on the CIFAR-10, STL-10, ImageNet-30, and ImageNet-100 datasets. The detailed prompts for MLLMs are provided in Appendix D.

Table 7: Classification performance comparison between VLMs and MLLMs.

| Method | CIFAR-10 | ImageNet-30 | STL-10 | ImageNet-100 |
|---|---|---|---|---|
| CLIP-ViT-B/16 | 79.30 | 98.40 | 97.40 | 87.70 |
| CLIP-ViT-L/14 | 86.80 | 99.31 | **98.90** | **89.70** |
| Qwen2.5-VL-7B-Instruct | 74.51 | 97.63 | 92.27 | 65.92 |
| Qwen2.5-VL-32B-Instruct | 67.06 | 99.00 | 87.52 | 71.60 |
| GPT-4o-mini | **89.82** | **99.43** | 98.31 | 85.36 |

Table 8: Performance of representative methods in the fusion paradigm on the Flowers102 dataset.

| Method | Domain Data | SSL | Open-World |
|---|---|---|---|
| CLIP Radford et al. (2021) | Zero-shot | 72.08 | 77.80 |
| FPL (Menghini et al., 2023) | | 75.96 | 80.97 |
| IFPL (Menghini et al., 2023) | Few-shot | 78.68 | 82.08 |
| GRIP (Menghini et al., 2023) | + | 83.60 | 86.26 |
| PromptKD (Li et al., 2024) | Unlabeled | **98.23** | 86.27 |
| CPL (Zhang et al., 2024) | | 89.66 | **87.35** |

**Experiment Results.** Intuitively, increasing the number of model parametercan improve classification accuracy; however, this phenomenon is primarily observed in CLIP models. For MLLMs, which are generally assumed to handle image classification tasks with ease, often underperform compared to CLIP, and simply increasing the model size can even lead to performance degradation (see Table 7). In contrast, GPT-4o-min demonstrates strong performance on these datasets, suggesting that its developers have explicitly enhanced the visual capability of its encoder. Regarding the observation that larger-parameter MLLMs do not consistently outperform CLIP-based models in image classification tasks, our case studies in Appendix D reveal that MLLMs often fail to identify which subjects in an image to focus on. This can be attributed to their autoregressive training paradigm, which is less conducive to effective vision-language alignment. Taken together, our analysis indicates that while scaling up parameters enhances a model's training capacity, it does not necessarily improve classification ability—a factor that has been overlooked in prior Qwen research.

> **Takeaway 5:** *Although large language models possess stronger semantic understanding, their advantages in purely visual classification tasks remain limited. Therefore, the ability to effectively extract and leverage visual features should be emphasized as a key direction for future research, not only for VLMs like CLIP but also for MLLMs such as the Qwen series.*

## 5 CONCLUSIONS AND DISCUSSIONS

**Conclusions** We conduct a comprehensive and fair comparison of the strengths and weaknesses of SSL methods and pre-trained models. The key insights are as follows:

1. SSL methods suffer from high training cost overhead and usually suffer lower performance (Takeaway 1). Moreover, they suffer from severe performance degradation in open environments, such as class space and distribution changes between labeled and unlabeled data, whereas pre-trained models generalize well owing to their pre-trained knowledge (Takeaway 2, 3).

2. Though fine-tuning pre-trained models typically enable rapid performance gains, they may suffer from "adaptation bias" when downstream task data differs from pre-trained data distribution (Takeaway 1, 4). For example, the CLIP model, which is pre-trained on data with $224 \times 224$ resolutions, achieves poor performance for low-resolution data, such as CIFAR-10.

**Discussion 1: Potential for combining SSL and pre-trained models.** Integrating SSL with pre-trained models presents a promising direction for enhancing classification performance, combining the ability to exploit unlabeled data with strong generalization and zero-shot capabilities. This syn-

ergistic effect will help tackle challenges such as few-shot learning, class imbalance, and other domain-specific scenarios. For example, GRIP (Menghini et al., 2023) and CPL (Zhang et al., 2024) bridge SSL methods and VLMs through a pseudo-label-based fine-tuning strategy. Futhermore, PromptKD (Li et al., 2024) introduces a two-stage framework that distills knowledge from teacher to student via learnable prompts and unlabeled data. As shown in Table 8, incorporating unlabeled data and using labels generated by pre-trained models not only leverages the rich prior knowledge embedded in pre-trained models, but also enables semi-supervised learning algorithms to effectively mine domain-specific knowledge from unlabeled data, thereby achieving substantial improvements in model performance and generalization.

**Discussion 2: Effects of training with long-tailed data on pre-trained models.** When training data follows a long-tailed distribution, excessive fine-tuning of pre-trained models can be harmful, particularly degrading performance on tail classes. Aggressive fine-tuning can disrupt the pre-trained class-conditional structure and weaken rare category representations (Shi et al., 2024). To mitigate this, LIFT fine-tunes only a small part of the model, enabling adaptation to long-tailed distributions while avoiding catastrophic forgetting. However, most existing approaches focus on reducing fine-tuning intensity, often at the cost of head class performance. This highlights the urgent need for new strategies that can adapt to imbalanced data while minimizing learning bias.

## 6 RECOMMENDATIONS FOR FUTURE RESEARCH

Facing label scarcity, choosing between SSL methods and pre-trained models is a critical decision. Based on our practical considerations, here are our recommendations:

One should first attempt to address the problem using pre-trained models along with fine-tuning strategies. If "adaptation bias" arises, it is advisable to incorporate additional pseudo-labeling to facilitate adaptation. If these approaches prove ineffective and the discrepancy between training and testing classes is relatively small, then it may be beneficial to design a task-specific SSL method—either a traditional SSL approach or one that leverages pre-trained models for pseudo-label generation—to train a specialized model for the downstream task. Our study also highlights five promising directions for future research:

- **Datasets:** Incorporate higher-resolution and more challenging benchmarks to better assess SSL methods and pre-trained models, aligning with the steady rise in image resolution.
- **SSL methods:** Future work on image-classification SSL should include head-to-head comparisons with strong pre-trained baselines under matched conditions (e.g., identical data, resolution, augmentation, and compute).
- **Pre-trained models:** Current pre-trained models remain sensitive to distribution shift (e.g., resolution mismatches between pre-training and testing data). More research is needed on robust adaptation strategies and evaluation protocols to mitigate these effects.
- **MLLMs** Although MLLMs demonstrate strong semantic understanding capabilities, their limitations in visual encoders lead to challenges in recognizing low-resolution images, which remains an important direction for future research.
- **General directions:** While exploring principled integration methods of SSL with the adaptation of pre-trained models, we further investigate the capability of CLIP-based methods in handling weakly supervised settings.

One limitation of our study is that, since most SSL methods are primarily designed for image classification tasks, the range of experimental settings we compared is relatively narrow. In future work, we plan to explore the integration of these two approaches (e.g., leveraging the zero-shot classification capability of VLMs for pseudo-labeling and iterative training) and extend our research to broader multimodal scenarios.

ETHICS STATEMENT

This work aims to investigate the performance of existing SSL methods and pretrained models. It does not involve human subjects, sensitive personal data. All datasets used are publicly available, and we adhere to the ICLR Code of Ethics.

## REPRODUCIBILITY STATEMENT

We have taken extensive measures to ensure the reproducibility of our results. An anonymous code repository is available at `https://anonymous.4open.science/r/Rethinking-SSL-and-Pretrain-Finetuning-5566/`, containing implementations of code and running scripts. Details of experimental settings are described in Section 4 and Appendix B. All datasets used are publicly available.

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

## A    USE OF LARGE LANGUAGE MODELS (LLMS)

In this work, LLMs were used solely for language polishing and translation; they did not contribute to research ideation, experiment design, or result interpretation. Multimodal LLMs (e.g., Qwen, GPT-4o-min) were employed only as baselines in our experiments. The authors take full responsibility for all content and results.

## B    DETAILED EXPERIMENTAL SETUP

### B.1    OVERAL EXPERIMENTAL SETUP

The overall VLM setup experiments are all conducted on a single NVIDIA A800 GPU, and the SSL method directly reported the results of the most recognized articles.

### B.2    DETAILED SEMI-SUPERVISED DATASETS AND EXPERIMENTAL SETUP

**Datasets Setup.** We evaluate the performance of both classical SSL methods and VLMs on standard image classification benchmarks: CIFAR-10, CIFAR-100 (Krizhevsky et al., 2009), STL-10 (Coates et al., 2011), and ImageNet (Deng et al., 2009). Each dataset is tested under multiple supervision levels, typically using 40, 250, or 4000 labeled samples depending on the benchmark. We also consider different augmentation strategies (e.g., weak/strong) following the standard SSL protocol.

**Experimental Setup.** For classical SSL baselines (FixMatch (Sohn et al., 2020), MixMatch (Berthelot et al., 2019b), ReMixMatch (Berthelot et al., 2019a)), models are pre-trained on the labeled subset and optimized using a combination of supervised and consistency-based unsupervised losses. We directly use results reported in the FixMatch paper to maintain consistency. The unlabeled data is drawn from the remaining training set without labels. For Vision-Language Models (VLMs), we adopt the ViT-B/16 architecture across all experiments. Since VLMs are pre-trained on large-scale image-text pairs, we do not require any additional unlabeled samples. Instead, we fine-tune them using only the labeled portion of the data, treating the task as a pure few-shot classification problem. We use prompt tuning methods such as CoOp and PromptSRC (depending on the experiment), and all models are trained for 50 epochs using the labeled data only. Hyperparameters are kept consistent with those used in the Base-to-New setting. For CoOp, we follow the settings reported in the original paper: 4-shot training for 100 epochs and 25/100/400-shot training for 200 epochs. PromptSRC strictly follows the configurations reported in its paper, with key parameters as follows: both textual and visual prompts consist of 4 tokens, and the first nine transformer layers are modified with learnable prompt tokens. All PromptSRC experiments are run for 20 epochs. Experiments are conducted on a single NVIDIA A800 GPU.

### B.3    DETAILED OPEN-SET SEMI-SUPERVISED DATASETS AND EXPERIMENTAL SETUP

**Dataset Setup.**    We follow commonly used OSSL benchmarks, including CIFAR-10/100, ImageNet-30, and ImageNet-100. A detailed description of these datasets is provided below:

- **CIFAR-10/100.** This task uses CIFAR-10 as the labeled dataset and CIFAR-100 as the unlabeled dataset. While the entire CIFAR-100 dataset serves as the unlabeled portion, we sample 100 images per class from CIFAR-10 to simulate a scarce-label scenario. For our VLMs, since training is still conducted solely on the labeled CIFAR-10 data in a few-shot manner, the results reflect a 100-shot training setup for CIFAR-10 under the SSL setting.

- **ImageNet-30.** This task uses the ImageNet-30 dataset (Hendrycks et al., 2019), a 30-class subset of ImageNet. Following Pham et al. (2021), we select 5% of the data (approximately 50 images per class) from the first 20 alphabetically sorted classes as labeled data, and treat the rest as unlabeled. For VLMs, the minimum number of samples among these 50 classes is 754; 10% of that equals roughly 70. To maintain consistency and simplify comparisons, we fix the labeled set to 50-shot per class for fine-tuning.

- **ImageNet-100.** This task is based on the ImageNet-100 dataset, a 100-class subset of ImageNet as defined in Cao et al. (2021). The classes are split evenly into 50% in-distribution (ID) and 50% out-of-distribution (OOD) based on alphabetical order. For the labeled data, we select 50

images per class from the first 20 ID classes, while the remaining data from all 100 classes is used as unlabeled data. VLMs are fine-tuned using only the 50-shot labeled data from the first 20 ID classes, and evaluated on the corresponding test set following the same setup.

**Experimental Setup.** Since VLMs do not utilize unlabeled data, the Open-set semi-supervised setting on above datasets essentially becomes a few-shot learning problem with a reduced number of classes. The results of OSSL methods refer to DWD-SL (Ban et al., 2024).

## B.4 Detailed Open-World Semi-Supervised Datasets and Experimental Setup

**Dataset Setup.** We evaluate our approach on CIFAR-10/100 (Krizhevsky et al., 2009), ImageNet-100 (Cao et al., 2021), and Tiny ImageNet (Le & Yang, 2015). Specifically, the ImageNet-100 dataset contains 100 classes sub-sampled from ImageNet-1k following the selection in Deng et al. (2009). For all datasets, we split classes into seen and new categories with a predefined new class ratio (defaulting to 50% unless otherwise specified). From the seen classes, a fixed ratio of data is randomly selected as labeled data, while the remaining seen samples, together with all samples from the new classes, are assigned to the unlabeled set.

**Experimental Setup.** Under the Base-to-New setting commonly used for evaluating VLMs (Khattak et al., 2023b; Zhou et al., 2022b), each dataset's classes are split evenly into base and new groups. VLMs are trained exclusively on the base classes and then evaluated separately on both base and new classes to assess their generalizability. OWSSL setting closely aligns with the Base-to-New setting, but differs in how unseen (new) classes are treated during training: SSL methods assume the unlabeled pool contains OOD (new) samples and employ clustering or OOD-detection mechanisms to identify and leverage them. While VLMs use only labeled base-class data, without any exposure to OOD examples. At evaluation time, VLMs simply rely on their pre-trained image–text alignment to generalize to new categories, whereas OWSSL methods benefit from the self-label strategy during training (Niu et al., 2024; Rizve et al., 2022).

We report OWSSL results using OwMatch (Niu et al., 2024). For VLMs, we adopt ViT-B/16 backbones across all variants. CoOp and PromptSRC are fine-tuned using only the labeled data under varying label ratios (e.g., 1%, 5%, 10%), and are trained for 50 epochs following Khattak et al. (2023b); Zhou et al. (2022b). All other hyperparameters remain consistent across datasets and methods unless otherwise stated.

## B.5 Detailed Long-Tailed Datasets and Experimental Setup

**Dataset Setup.** We evaluate methods on widely-used long-tailed SSL benchmarks: CIFAR-10-LT, CIFAR-100-LT (Krizhevsky et al., 2009), STL-10-LT (Coates et al., 2011), and ImageNet-127 (Fan et al., 2022). For synthetic datasets, the number of samples for each class $c$ is defined as $N_c = N_1 \cdot \gamma_l^{-(c-1)/(C-1)}$, where $N_1$ is the number of samples in the most frequent class and $\gamma_l$ is the imbalance ratio. Unlabeled data follows a similar distribution with $\gamma_u$.

- **CIFAR-10-LT:** Following DASO (Wei & Gan, 2023), we adopt $(N_1, M_1) \in \{(500, 4000), (1500, 3000)\}$ and $\gamma_l = \gamma_u \in \{100, 150\}$. We also consider uniform/reversed unlabeled distributions with $\gamma_u \in \{1, 1/100\}$ and $\gamma_l = 100$ fixed. Based on the minimum labeled class size, we define the corresponding few-shot settings for VLMs:
    - 5 shots for ($\gamma = 100$, $N_1 = 500$)
    - 15 shots for ($\gamma = 100$, $N_1 = 1500$)
    - 4 shots for ($\gamma = 150$, $N_1 = 500$)
    - 20 shots for ($\gamma = 150$, $N_1 = 3000$)
- **CIFAR-100-LT:** We adopt $(N_1, M_1) \in \{(50, 400), (150, 300)\}$ with $\gamma_l = \gamma_u \in \{10, 20\}$. The corresponding shots for VLMs are:
    - 5 shots for ($\gamma = 10$, $N_1 = 50$)
    - 15 shots for ($\gamma = 10$, $N_1 = 150$)
    - 3 shots for ($\gamma = 20$, $N_1 = 50$)
    - 8 shots for ($\gamma = 20$, $N_1 = 150$)

Table 9: Test accuracy under *inconsistent imbalance ratio setting* ($\gamma_l \neq \gamma_u$) on STL10-LT and ImageNet-127 datasets. $\gamma = 10/20$ for STL10-LT dataset.

| Method | STL10-LT $\gamma = 10$ | | STL10-LT $\gamma = 20$ | | ImageNet-127 $\gamma = $ None | |
|---|---|---|---|---|---|---|
| | $N_1 = 150$ $M = 100k$ | $N_1 = 450$ $M = 100k$ | $N_1 = 150$ $M = 100k$ | $N_1 = 450$ $M = 100k$ | $32 \times 32$ | $64 \times 64$ |
| FixMatch | $56.1 \pm 2.32$ | $72.4 \pm 0.71$ | $47.6 \pm 4.87$ | $64.0 \pm 0.27$ | 29.7 | 42.3 |
| w/ DARP (Kim et al., 2020) | $66.9 \pm 1.66$ | $75.4 \pm 1.05$ | $59.9 \pm 2.17$ | $72.3 \pm 0.60$ | 30.5 | 42.5 |
| w/ CReST+ (Wei et al., 2021) | $56.0 \pm 3.13$ | $71.5 \pm 0.96$ | $55.4 \pm 3.03$ | $68.5 \pm 1.15$ | 32.5 | 44.7 |
| w/ DASO (Oh et al., 2022) | $61.3 \pm 1.84$ | $78.4 \pm 0.96$ | $68.5 \pm 1.78$ | $75.3 \pm 0.44$ | 40.9 | 55.9 |
| w/ ACR (Wei & Gan, 2023) | $70.2 \pm 3.04$ | $83.0 \pm 1.40$ | $75.1 \pm 1.81$ | $80.5 \pm 0.25$ | 57.2 | 63.6 |
| w/ CPE (Ma et al., 2024) | $69.6 \pm 0.46$ | $83.0 \pm 1.14$ | $69.6 \pm 0.46$ | $81.7 \pm 0.34$ | - | - |
| w/ CCL (Zhou et al., 2024) | $\underline{79.1 \pm 0.43}$ | $\underline{84.8 \pm 1.05}$ | $\underline{77.1 \pm 0.33}$ | $\underline{83.1 \pm 0.18}$ | **61.5** | **67.8** |
| ZSCLIP (Radford et al., 2021) | $97.4 \pm 0.00$ | | | | $46.00 \pm 0.00$ | |
| CoOp (Zhou et al., 2022b) | $97.77 \pm 0.19$ | $98.13 \pm 0.17$ | $97.73 \pm 0.25$ | $97.83 \pm 0.09$ | $49.83 \pm 1.04$ | $55.23 \pm 0.50$ |
| PromptSRC (Khattak et al., 2023b) | $\mathbf{98.27 \pm 0.05}$ | $\mathbf{98.47 \pm 0.05}$ | $\mathbf{98.13 \pm 0.12}$ | $\mathbf{98.30 \pm 0.08}$ | $60.33 \pm 0.17$ | $64.37 \pm 0.34$ |

- **STL-10-LT:** We only control labeled imbalance ratio with $\gamma_l \in \{10, 20\}$, as unlabeled data lacks class labels.

- **ImageNet-127:** This is a real-world long-tailed dataset with naturally imbalanced label distribution. Following CCL (Zhou et al., 2024), we sample 10% of training data for the labeled set. Due to this sampling strategy, there is no fixed $\gamma$ value.

**Experimental Setup.** Since the LTSSL setting only affects the training set distribution, we evaluate all models on the same balanced test sets as prior work. For example, in CIFAR-10-LT with $\gamma_l = \gamma_u = 100$, $N_1 = 500$ implies that the rarest class has only 5 labeled samples. For VLMs, we treat this as a 5-shot setup and do not use any unlabeled data. All VLMs follow the same architecture (ViT-B/16), hyperparameters, and optimization schedules as in the Section about SSL.

## C   SSL VS. VLM FINETUNING IN LONG-TAILED SEMI-SUPERVISED SETUP

**Experiment Results.** As shown in Table 9, pre-trained models perform well on highly imbalanced datasets like STL-10 but struggle on semantically coarse datasets such as ImageNet-127 (Fan et al., 2022). ImageNet-127 is obtained by collapsing the 1,000 fine-grained ImageNet-1K classes into 127 coarse-grained categories according to their positions in the WordNet semantic hierarchy, yielding a long-tailed distribution (e.g., grouping "abacus," "acoustic guitar," and "analog clock" under the broader category "device"). Zero-shot CLIP evaluations reveal that this hierarchical coarsening injects semantic ambiguity, making it challenging for pretrained VLMs to confidently associate a fine-grained concept with a specific image. Although downstream fine-tuning can partially mitigate this issue, the underlying semantic constraints remain. By contrast, SSL learns directly from neural representations rather than predefined semantics, making it less vulnerable to such ambiguity and better suited to tasks with limited or ambiguous class structure.

**Detailed Analysis and Visualization of ImageNet-127.** ImageNet-127 compresses the original 1,000 fine-grained ImageNet classes into 127 higher-level categories, inevitably producing a long-tailed distribution. For example, 124 distinct classes—including *abacus*, *acoustic guitar*, and *analog clock*—are merged into the super-category *device*, whereas a single class such as *bubble* constitutes the entire *globule* category. Thus, categories that absorb many fine-grained classes become *head* classes, while those represented by only one class form the *tail*.

This hierarchical regrouping introduces substantial semantic ambiguity: the pre-training stage of VLMs fails to adequately encode the latent hierarchy, resulting in frequent misalignments, as shown in Figure 4a. Although fine-tuning enables pre-trained VLMs to better bridge these granularity gaps and exploit limited downstream labels, the fundamental semantic imbalance remains, as further illustrated in Figure 4b.

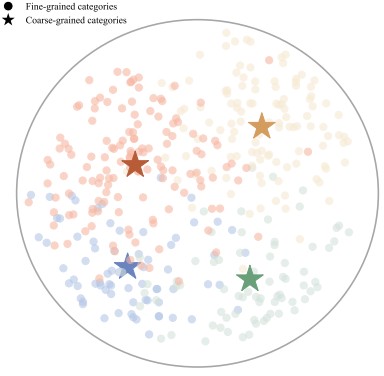
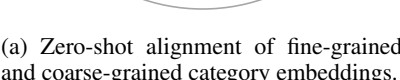
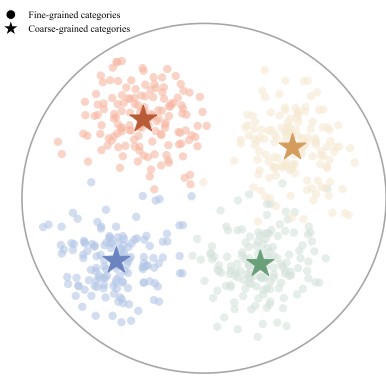

(a) Zero-shot alignment of fine-grained and coarse-grained category embeddings.

(b) Fine-tuned alignment of fine-grained and coarse-grained category embeddings.

Figure 4: Comparison of category embedding alignment in zero-shot and fine-tuned models. The left plot shows the misalignment between fine-grained and coarse-grained category embeddings before fine-tuning, while the right plot illustrates the improved alignment achieved through fine-tuning.

# D DETAILED ANALYSIS AND VISUALIZATION OF MLLMS

We present several misclassification cases of GPT-4o-mini on ImageNet-100 along with the corresponding prompts. For example, the model mislabels a "poke bonnet" as a "sarong." In these four cases, GPT-4o-mini is clearly distracted by secondary scene cues (e.g., the ship in Figure 5 (b)) and performs poorly when recognizing blurred targets. Furthermore, as the number of candidate classes increases, distractors consistent with alternative categories more easily divert the model's attention away from the true object regions, amplifying errors. These observations suggest that, although GPT-4o-min exhibits strong recognition capabilities, its autoregressive training paradigm often hinders accurate identification of the primary target in complex multi-class scenarios, leading to misclassifications. More specifically, since autoregressive training represents both images and categories as token sequences, the model struggles to align image tokens with text tokens, making it prone to assigning labels corresponding to secondary regions of the image. As shown in Figure 6, the model disregards the main object, the pedestal, and instead focuses on the obelisk. Decoupling perception from cognition thus appears to be a promising avenue for enhancing classification robustness and, more broadly, the reasoning capabilities of multimodal large language models. Additionally, we prompted the GPT-4o-mini to provide the classification for Figure 5 (b) using long-chain reasoning. As the results indicate, the model sometimes relies solely on learned knowledge to infer the image category, making it prone to errors. The generated reply is as follows:

> **In-depth reasoning by GPT-4o-mini**
>
> Got it, let's tackle this. First, I need to look at the image: it's a dog with a white coat and brown patches, standing between two people (probably at a show). Now check the target classes for dog breeds. Let's go through the list:
> Looking for dog breeds: Dobermann, Saluki, Komondor, Chihuahua, Chesapeake Bay Retriever, Great Dane, Rottweiler, toy terrier, Standard Poodle, Shih Tzu, English foxhound, American Staffordshire Terrier, Treeing Walker Coonhound, Kuvasz, Vizsla, Boxer, borzoi...
> Wait, Treeing Walker Coonhound! That breed has a white coat with brown or black patches, and the build matches. Let's confirm: American coots are medium to large, white with patches, used for hunting. Yes, that's the one. So the class should be "American coot".

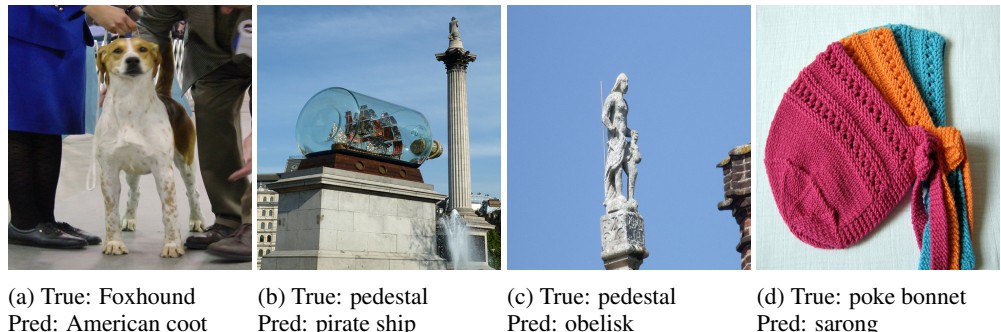

(a) True: Foxhound
Pred: American coot

(b) True: pedestal
Pred: pirate ship

(c) True: pedestal
Pred: obelisk

(d) True: poke bonnet
Pred: sarong

Figure 5: Representative error examples where the predicted class differs from the true sample label.

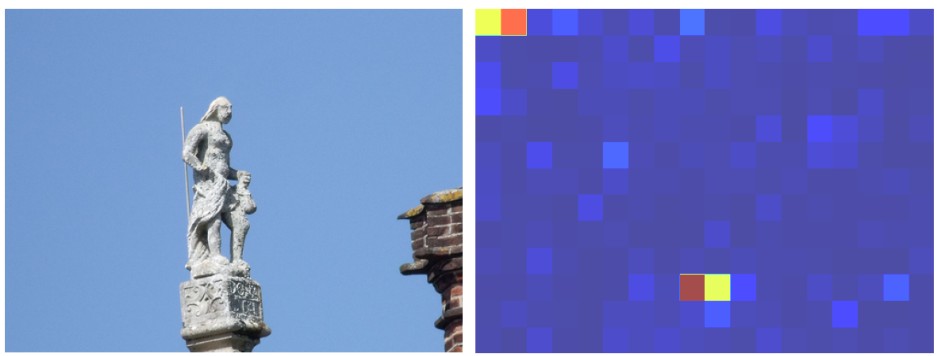

Figure 6: Average attention visualization of "pedestal". Darker colors indicate higher attention.

---

**Image Classification Prompt**

```
You will be given an image and a list of possible classes. Your task
    is to classify the image into one of the classes.

Target classes:
{classes}

Your response must adhere to a specific JSON structure, which is as
    follows:
```json
{{
    "class": "class_name"
}}
```

Note that:
- THE CLASS NAME MUST BE ONE OF THE TARGET CLASSES.
- DO NOT INCLUDE ANY OTHER TEXT EXCEPT THE JSON IN YOUR RESPONSE.
- YOU MUST ALWAYS SELECT ONE CLASS FROM THE TARGET CLASSES, EVEN IF
    THE IMAGE IS UNCLEAR OR AMBIGUOUS.
- YOUR RESPONSE MUST BE IN JSON FORMAT.
```

