# OpenReview forum: "Unlabeled Data vs. Pre-trained Knowledge: Rethinking SSL in the Era of Large Models"
_ICLR.cc/2026/Conference — Submitted to ICLR 2026_

### Official Review · Reviewer_bHUs · 2025-10-15

**Soundness:** 4
**Presentation:** 4
**Contribution:** 3
**Rating:** 8
**Confidence:** 4

**Summary:**

This paper compares unlabeled data and pre-trained models with a limited amount of labeled data. Various semi-supervised learning scenarios were considered, including semi-, open-set, open-world, and long-tailed semi-supervised learning. Based on extensive experimental results, valuable conclusions have been drawn, showing that pre-trained models can often perform better, though they still face many limitations and require further research.

**Strengths:**

- The problem studied, i.e., comparing unlabeled data and pre-trained models, is interesting and important to the literature.

- The experiments are comprehensive and cover various semi-supervised learning settings, including standard, open-set, open-world, and long-tailed semi-supervised learning. Therefore, the conclusions are convincing.

- The writing is excellent, making the paper easy to understand.

- This paper is an important position paper for the field of weakly supervised learning.

**Weaknesses:**

- There is a lack of related work discussing the use of zero-shot CLIP for traditional weakly supervised learning, a more general form of semi-supervised learning. Additionally, some work discusses semi-supervised learning with CLIP.

- The title would be better with "semi-supervised learning" instead of "SSL" since "SSL" has multiple meanings, such as self-supervised learning.

- In experiments on standard semi-supervised learning, SOTA SSL algorithms such as FixMatch seem to achieve better performance on CIFAR-10 and CIFAR-100. Therefore, concluding that pre-trained models are better for standard semi-supervised learning is incomplete.

[1] Delving into Weakly Supervised Learning with Pre-Trained Models, https://openreview.net/forum?id=RgWATMmWmz

**Questions:**

- For open-set and open-world SSL learning, is the method using pre-trained models the same as standard semi-supervised learning since it does not use unlabeled data?

---

> ### Author Response · Authors · 2025-11-16
>
> We sincerely thank you for the very positive and encouraging assessment of our work.
>
> **W1: Related work on using zero-shot CLIP for traditional weakly supervised learning is missing.  Additionally, some work discusses semi-supervised learning with CLIP.**
>
> A1: Thank you for your suggestion. In our analysis, we focus on the strengths and weaknesses of semi-supervised learning methods and pre-trained models across various semi-supervised scenarios, while further exploring the potential of their combination. The application of zero-shot CLIP to broader settings such as weakly supervised learning is also worthy of in-depth study. However, due to space constraints, we will provide a preliminary discussion in the **RECOMMENDATIONS FOR FUTURE RESEARCH** section and encourage researchers to conduct further exploration.
>
> Additionally, existing works ([1], [2], [3]) have discussed methodological integrations of semi-supervised learning and CLIP, but they lack in-depth, multi-angle analysis of the respective advantages and disadvantages of the two paradigms. Our work aims to systematically summarize the research directions of semi-supervised learning in the pre-trained model era, provide guidance for subsequent researchers on method selection across different scenarios, and further call for research on in-depth integration paradigms.
>
> [1] Candidate pseudolabel learning: enhancing vision-language models by prompt tuning with unlabeled data
>
> [2] Enhancing CLIP with CLIP: Exploring Pseudolabeling for Limited-Label Prompt Tuning
>
> [3] PromptKD: Unsupervised Prompt Distillation for Vision-Language Models
>
> **W2: Some studies on semi-supervised learning with CLIP are not discussed.**
>
> A2: Thank you for the comment. However, **in Discussion 1**, we explicitly discuss the research direction of combining semi-supervised learning methods with VLMs. In particular, we highlight representative works such as GRIP, CPL, and PromptKD, and explain how these approaches leverage pseudo-label refinement and prompt-based knowledge distillation to fuse the prior knowledge of pre-trained VLMs with domain information from unlabeled data. This line of work has been shown to be effective in improving classification performance.
>
> **W3: The title would be better with "semi-supervised learning" instead of "SSL" since "SSL" has multiple meanings, such as self-supervised learning.**
>
> A3: We thank you for pointing out this issue, and we will revise the ambiguous phrasing in the updated version of the paper.
>
> **W4: The experiments overlook that SOTA SSL methods like FixMatch outperform pre-trained models on CIFAR-10/100, making the conclusion about pre-trained models’ superiority incomplete.**
>
> A4: We thank you for this comment. In Section 4.2, we analyzed the disadvantages of pre-trained VLMs, and concluded that VLMs adapt efficiently but are more susceptible to adaptation bias. We will revise Takeaway 1 in the updated version to explicitly emphasize the advantage of semi-supervised learning when dealing with low-resolution images.
>
> **Q1: For open-set and open-world SSL learning, is the method using pre-trained models the same as standard semi-supervised learning since it does not use unlabeled data?**
>
> A1: Your understanding is correct. In our setting, the VLM-based methods for open-set and open-world semi-supervised settings  use the same few-shot training strategy as in the standard setting, and they do not exploit unlabeled data. This design is intended to compare pretrained VLMs with semi-supervised learning under a small labeled-data budget.

---

> > ### Comment · Reviewer_bHUs · 2025-11-26
> >
> > Thanks for the reply, that address my concerns. I will maintain my score.

---

### Official Review · Reviewer_c4Ht · 2025-11-01

**Soundness:** 3
**Presentation:** 3
**Contribution:** 3
**Rating:** 4
**Confidence:** 3

**Summary:**

The paper presents a controlled comparison of classic semi-supervised learning (SSL) methods versus pre-trained vision–language models (VLMs) across four regimes: standard SSL, open-set, open-world, and long-tailed classification. Under matched supervision budgets, the study finds that VLMs (e.g., CLIP with prompt/adapter tuning) often match or surpass SSL in accuracy while being more compute-efficient.

**Strengths:**

This is a timely study in the era of large models and has the potential to deliver useful insight for the SSL community. The experiments are relatively comprehensive, covering main settings in SSL research. Takeaways are clearly stated.

**Weaknesses:**

The benchmark datasets—CIFAR-10/100, STL-10, and ImageNet—may overlap semantically with VLM pretraining data, potentially advantaging the VLMs in this comparison. By contrast, in domains where the pre-trained model has seen less data (e.g., medical imaging), it is unclear whether the conclusions would still hold. Experiments probing this low-overlap setting are currently missing.

**Questions:**

NA

---

> ### Author Response · Authors · 2025-11-16
>
> We thank you for acknowledging the breadth of our experiments and for recognizing the insightful takeaways drawn from them.
>
> **W1: Exploratory experiments on low-overlap scenarios (e.g., medical scenarios) are lacking.**
>
> A1: We thank you for pointing out this issue. We have indeed considered the impact of distribution mismatch between the pre-training data and downstream tasks on the performance of pretrained models, and we analyze this phenomenon through the lens of **adaptation bias** in the main paper. However, since the exact composition of CLIP's pre-training corpus is not publicly available, we cannot precisely quantify overlap for arbitrary downstream datasets. Instead, we choose a setting where the overlap is plausibly very low: medical imaging. Following [1], we evaluate several semi-supervised learning methods and VLMs on a medical image classification dataset. The results are:
>
> | Method           | Accuracy       | Sensitivity | Specificity | F1                |
> |------------------|----------------|-------------------|-------------------|-------------------|
> | Self-training    | 92.37         | 67.63            | **93.31**             | 54.51             |
> | FlexMatch  | **93.40±0.05**     |71.25±0.23        | 92.23±0.17        | 61.37±0.22        |
> | CoMatch | 93.39±0.04     | 70.22±0.25        | 92.29±0.11        | **61.94±0.21**        |
> | FixMatch | 93.23±0.14     | **72.91±0.52**        | 92.21±0.03        | 60.19±0.31        |
> | Zero-shot CLIP   | 14.60          | 22.40             | 86.40             | 11.10             |
> | CoOp             | 55.43±2.49     | 52.63±2.32        | 92.33±0.50        | 41.80±2.01        |
> | PromptSRC        | 59.97±2.09     | 60.70±2.87        | 93.17±0.37        | 51.10±1.77        |
>
> Even after fine-tuning, VLMs remain far from semi-supervised learning on this dataset. This indicates that adaptation bias continues to affect pretrained models in domains that are under-represented in their pre-training data, and that fine-tuning alone is insufficient to close the gap. These results further support our main claim that the impressive generalization of pretrained models is inherently **conditional** on the alignment between pre-training and downstream distributions, and that semi-supervised learning remains a strong and sometimes preferable alternative when this alignment is weak.
>
> [1] FixMatch-LS: Semi-supervised skin lesion classification with label smoothing

---

> ### Author Response · Authors · 2025-11-27
> **Official Comment by Authors**
>
> Dear Reviewer c4Ht,
>
> Thank you very much for your valuable comments during the review process. We have carefully answered your concerns by providing additional explanations and supporting experimental evidence in our response.
>
> We sincerely hope that our reply adequately resolves your questions. If you have further inquiries or suggestions, we would be more than happy to continue the discussion and provide any additional information needed.
> Thank you again for your time and effort in reviewing our work.
>
> Best regards,
>
> The Authors of Submission 7026

---

### Official Review · Reviewer_Gqrj · 2025-11-01

**Soundness:** 3
**Presentation:** 3
**Contribution:** 2
**Rating:** 2
**Confidence:** 5

**Summary:**

This paper investigates traditional self-supervised learning (SSL) methods and CLIP-based prompt-tuning approaches. The authors conduct experiments across various image classification settings and present several empirical findings.

**Strengths:**

The paper presents comprehensive experiments.

**Weaknesses:**

1. This paper focuses solely on experimental comparisons without proposing any new method. Therefore, the experimental setup is critical and should be described in detail in the main text.

2. The comparison settings are entirely unfair. For example, traditional SSL methods use ResNet backbones (e.g., ResNet-28-2), whereas prompt-tuning methods use ViT/B-16. Moreover, comparing SSL and few-shot learning is conceptually unsound. SSL with CLIP has already been explored, and under such unfair experimental conditions, the conclusions drawn from the experiments are not meaningful.

3. Since CLIP is trained on diverse web-scale data, its robustness to OOD (out-of-distribution) data is expected and not surprising.

4. Recent studies [1] have shown that MLLMs perform better in image classification tasks using no-thinking or direct-answering prompts. It would be more informative to report the performance of MLLMs under such settings, as this would better reflect their true classification capabilities.

5. CLIP models can also be used for SSL settings. For example, [2]. It is questionable why comparing few-shot CLIP-based prompting methods with ResNet-based SSL methods.

6. Overall, the contribution of this paper appears limited. The authors merely conduct inference-based experiments and report results that are largely unsurprising. There are no technical or methodological innovations.

Reference:
[1] To Think or Not To Think: A Study of Thinking in Rule-Based Visual Reinforcement Fine-Tuning. NeurIPS 2025.

[2] Candidate pseudolabel learning: enhancing visionlanguage models by prompt tuning with unlabeled data. ICML 2024.

**Questions:**

See weaknesses.

---

> ### Author Response · Authors · 2025-11-16
>
> We thank your for the careful reading and constructive comments. Our paper is intentionally an empirical study rather than a method paper. The main goal is to answer a practical question that many practitioners face in the “foundation model era”: given the same limited amount of labeled data, when should we prefer (i) classic semi-supervised learning from scratch, (ii) pre-trained models, (iii) their combinations? Our contributions are a systematic comparison under carefully controlled label budgets and a set of boundary conditions that clarify where each paradigm is suitable or fails.
>
> **W1: The experimental setup is briefly described in the main text.**
>
> A1: We thank you for pointing out this issue. Due to space constraints, in the main text we focus on the key design choices of our experimental protocol, in particular the strict control of the labeled data budget across all methods to ensure fair comparison. The remaining implementation details are therefore deferred to the appendix. This allows us to devote the main body of the paper to a systematic analysis of how semi-supervised learning and pre-trained models behave under different settings, and to distill the resulting practically useful conclusions.
>
> **W2.1: Different model architectures lead to unfair experimental settings**
>
> A2.1: Thank you for the suggestion. We would like to clarify that different tasks and learning paradigms naturally favor different backbone architectures. In semi-supervised learning, it is standard practice to use ResNet-28 variants on low-resolution datasets such as CIFAR-10/100 and ResNet-50 on ImageNet-scale datasets ([1], [2], [3]). Our choice to report FixMatch with an RN28 backbone on CIFAR therefore follows established benchmarks, and we will make this explicit in the revised description of the experimental setup. Importantly, our paper also analyzes how architecture and image resolution jointly affect performance (see Figure 2/3). CLIP employs increasingly complex backbones (from RN50 to ViT-B/16) in order to leverage modern architectures for extracting visual information. However, due to the distribution mismatch between pre-training data and low-resolution downstream datasets, its performance on CIFAR-like benchmarks remains inferior to semi-supervised learning methods despite its stronger backbone. This contrast further reinforces our main message: practitioners should choose different paradigms depending on data characteristics, rather than assuming that larger pre-trained models are always preferable.
>
> [1] OwMatch: Conditional Self-Labeling with Consistency for Open-World Semi-Supervised Learning
>
> [2] FixMatch: Simplifying Semi-Supervised Learning with Consistency and Confidence
>
> [3] ReMixMatch: Semi-Supervised Learning with Distribution Alignment and Augmentation Anchoring
>
> **W2.2: Unsound comparison and meaningless conclusions.**
>
> A2.2: We have to emphasize that our work is specifically motivated by understanding when each paradigm should be preferred under a fixed labeled data budget, rather than by advocating any single paradigm as universally superior. To this end, we conduct a large number of fair comparisons (same amount of labeled data, best-practice backbone for each method) across classic semi-supervised learning, pre-trained models and their hybrid variants. Our main takeaways are:
> - no method is optimal in all scenarios;
> - when there is strong data bias such as very low image resolution, semi-supervised learning is generally the most suitable choice;
> - when rapid adaptation is required or images are high-resolution, few-shot tuning of pre-trained models is preferable;
> - when peak performance is required and computational time is less critical, combined approaches are the most attractive option.
>
> In this sense, our study plays a bridging role: it systematically compares semi-supervised learning and pre-trained models under diverse settings and datasets, and then discusses their respective usage regimes and future development directions. In practice, many researchers face the concrete question of which paradigm is more likely to yield strong or even SOTA performance under their specific constraints; our results are intended to provide precisely such guidance. Moreover, we argue that explicitly comparing semi-supervised learning and pre-trained models is particularly timely in the current era of large-scale pretraining, as it helps the community re-examine the limitations of both sides and motivates combined approaches that leverage their complementary strengths to further improve downstream performance.

---

> ### Author Response · Authors · 2025-11-16
>
> **W3: CLIP’s robustness to OOD data is expected and unsurprising given its web-scale training.**
>
> A3: We agree with you that CLIP's strong robustness is largely due to its web-scale and highly diverse pre-training data, and this is indeed a key reason why many pre-trained models exhibit good OOD performance. However, our empirical findings suggest that this robustness is **conditional**: it depends on the downstream data being reasonably aligned with the pre-training distribution. When there is a strong mismatch, pre-trained models can suffer from what we refer to as **adaptation bias**. We further evaluate semi-supervised learning and VLMs on a medical imaging dataset, which lies far from CLIP's web pre-training distribution. The results are:
>
> | Method           | Accuracy       | Sensitivity | Specificity | F1                |
> |------------------|----------------|-------------------|-------------------|-------------------|
> | Self-training    | 92.37         | 67.63            | **93.31**             | 54.51             |
> | FlexMatch  | **93.40±0.05**     |71.25±0.23        | 92.23±0.17        | 61.37±0.22        |
> | CoMatch | 93.39±0.04     | 70.22±0.25        | 92.29±0.11        | **61.94±0.21**        |
> | FixMatch | 93.23±0.14     | **72.91±0.52**        | 92.21±0.03        | 60.19±0.31        |
> | Zero-shot CLIP   | 14.60          | 22.40             | 86.40             | 11.10             |
> | CoOp             | 55.43±2.49     | 52.63±2.32        | 92.33±0.50        | 41.80±2.01        |
> | PromptSRC        | 59.97±2.09     | 60.70±2.87        | 93.17±0.37        | 51.10±1.77        |
>
> Even after tuning, CLIP-based methods remain far behind semi-supervised learning on this dataset. This shows that the impressive generalization of pre-trained models does not transfer to all OOD domains, especially highly specialized ones like medical imaging. Thus, CLIP's robustness is powerful but inherently conditional, and they further motivate our main message: practitioners should carefully consider data characteristics when deciding between semi-supervised learning and pre-trained models.
>
> **W4: SSL with CLIP has already been explored.**
>
> A4: We would like to clarify that prior work combining semi-supervised learning with CLIP does not conflict with our contributions. Our goal is to characterize the bottlenecks and applicability regimes of different paradigms under a fair setting. Even though SSL+CLIP approaches have been explored, this paradigm is still not universally suitable across all scenarios (see A.2.2).
>
> **W5: The paper should report MLLM performance with no-thinking or direct-answering prompts to better reflect their true classification capabilities.**
>
> A5: We recoginize that you may have overlooked Table 6, where we explicitly evaluate MLLMs under a direct-answer setting. The key conclusion from this experiment is that, although MLLMs possess stronger semantic understanding, their advantages in purely visual classification tasks remain limited. Therefore, we argue that the ability to effectively extract and leverage visual features should be emphasized as a key direction for future research, not only for VLMs such as CLIP but also for MLLMs such as the Qwen series.

---

> ### Comment · Reviewer_Gqrj · 2025-11-19
> **Thank you for your response**
>
> Thank you for the detailed rebuttal. However, most of my main concerns are not addressed. Specifically,
>
> 1. The response does not justify the practical value of the comparison. Why would a practitioner choose Few-Shot Learning (FSL) if abundant unlabeled data is available? Since SSL is included as a baseline, unlabeled data clearly exists. In this scenario, ignoring unlabeled data to perform pure FSL is irrational.
>
> 2. Comparing methods across vastly different architectures (e.g., SSL on ResNet-28 vs. Pre-training on ViT-B) introduces confounding variables. It is impossible to attribute performance differences to the "paradigm" when model capacity differs so drastically. A scientific comparison must control for computational budget and backbone size. The reason that traditional SSL algorithms used ResNet is that there are no ViT that time.
>
> 3. The scope of "pre-trained models" is too narrow in this paper which only include CLIP. It should include self-supervised ViTs (e.g., MAE/DINO) and also CLIP ViTs. All these backbones can be used for SSL, e.g., with visual prompt tuning or other fine-tuning strategies. For example, in Reviewer bHUs's example paper, the performance of all weakly-supervised algorithms hugely boost when applying CLIP ViTs with Visual prompt tuning.
>
> 4.  Invalid CLIP Implementation. The poor performance of CLIP on low-resolution data appears to be an implementation flaw. CLIP (ViT-B) is trained on $224 \times 224$ images; applying it to native CIFAR resolution without upscaling is invalid. This performance drop is due to incorrect input preprocessing, not intrinsic "adaptation bias."
>
> 5. My concern regarding W5 has been addressed. However, the experimental setting should still be included in the main text. Presenting the MLLM setup only in the Appendix is not appropriate for a comparison-focused paper, where such details are essential.
>
> Overall, the current experimental results lack the robustness and depth required to provide actionable insights for practitioners.

---

> > ### Comment · Reviewer_Gqrj · 2025-11-19
> >
> > Additionally, the performance of CLIP-ViT-B16 is reported as 91.6 and 68.7 on the CIFAR-10 and CIFAR-100 datasets, respectively (see Table 11 in [1]). In this paper, however, the reported results are only 79.30 and 46.00. Authors should clarify the reason for such performance gap.
> >
> >
> > [1] Learning Transferable Visual Models From Natural Language Supervision

---

> > ### Author Response · Authors · 2025-11-21
> > **Official Comment by Authors**
> >
> > Thank you again for your constructive comments on our work.
> >
> > **W1: The actual value of the article is unclear**
> >
> > A: We can consider the following practical scenario: In a classification task, researchers only have access to a small amount of high-quality labeled data. They face two options: either **collect additional unlabeled data at a certain cost and conduct SSL with substantial time investment**, or **directly fine-tune a pre-trained model using the limited labeled data**. However, the optimal choice between these two approaches remains unclear.
> >
> > To answer this critical question, we investigate the performance, training efficiency, and other key metrics of both paradigms. For comprehensive exploration, we also evaluate the performance of methods combining CLIP with SSL. Our experimental conclusions are as follows:
> > *No single method is universally effective across all scenarios. In most cases, directly using a pre-trained model achieves the optimal balance between efficiency and performance. However, when choosing this approach, the data distribution of the target task must be considered—if it deviates significantly from the common pre-training data distribution, satisfactory results are difficult to obtain directly. In such cases, the only viable option is to collect more unlabeled data and adopt time-consuming SSL for training. Additionally, if large-scale unlabeled data can be easily collected and training time is not a critical constraint, combining pre-trained models with SSL emerges as a superior choice.*
> >
> > **W2: Comparing methods across vastly different architectures introduces confounding variables**
> >
> > A: To address your concern that architectural differences may introduce confounding variables, we conducted comparative experiments on FixMatch [1] with **ResNet-28 and ViT-B/16**, strictly following the standard settings in the original FixMatch paper (batch_size = 64, $\mu$ = 7).
> >
> > Our experimental results reveal that **the enormous difference in computational** overhead is the primary bottleneck limiting the adoption of ViT in SSL: **RN28-FixMatch only requires ~1GB of GPU memory, while ViT-B/16-FixMatch** **experiences** **an explosive increase to ~50GB**. This is because SSL inherently requires massive unlabeled data (μ=7 times the amount of labeled data) and applies both weak and strong augmentations to them. Even with a small batch size, the quadratic computational complexity of the self-attention mechanism in ViT leads to a prohibitive surge in memory demand. In contrast, **the ResNet architecture, due to its lightweight design, achieves highly competitive performance with extremely low resource consumption, making it still the most practically useful choice in the SSL field**. VLM methods such as CoOp benefit from their strong few-shot adaptation capabilities and do not rely on massive unlabled data. Thus, they can directly leverage the powerful representational capacity of ViT-B/16 at a low resource cost (~1GB as well) to achieve excellent classification performance.
> >
> > In summary, our choice of different backbones for different paradigms (SSL vs. pre-trained models) is the result of a comprehensive trade-off between model performance and practical deployment constraints. This ensures that we compare the two paradigms under their "best practice" configurations, enabling a meaningful and fair comparison from multiple perspectives at a higher level, rather than enforcing simplistic variable control that ignores real-world application bottlenecks. Furthermore, the original FixMatch [1] paper used wider networks (WRN-28-8) instead of deeper ones (ResNet-50) for CIFAR-100, indicating that increasing network depth does not contribute to performance improvement.
> >
> > [1] FixMatch: Simplifying Semi-Supervised Learning  with Consistency and Confidence. NeurIPS 2020.
> >
> > **W3: The scope of 'pre-trained models' is too narrow, including only CLIP**
> >
> > A: The core reason for selecting CLIP as the pre-trained model lies in **its maturity as a cross-modal pre-training framework—its fine-tuning techniques for downstream image classification tasks are well-established.**
> >
> > It is important to clarify that MAE and DINO are not inherently designed for direct application in image classification: MAE focuses on pixel-level reconstruction, prioritizing the learning of low-level structural and detailed features of images; DINO leverages self-distillation to achieve feature-level alignment, excelling at extracting highly discriminative general visual representations. However, **both models require additional fine-tuning strategies (e.g., adding classification heads, Visual Prompt Tuning) to adapt to classification tasks**. Furthermore, **the weak supervision paper mentioned by Reviewer bHUs also exclusively adopts CLIP, which underscores its representative status in related research**. We will further explore the performance of pre-trained models beyond CLIP under few-shot settings in future work.

---

> ### Author Response · Authors · 2025-11-21
> **Official Comment by Authors**
>
> **W4: Response to "CLIP Implementation during Testing and Reproducibility Performance Differences"**
>
> We aim to clarify a misunderstanding regarding our implementation. **All images input to the VLM undergo standard preprocessing, including upscaling low-resolution images (e.g., $32\times32$ CIFAR images) to the required input resolution of the CLIP image encoder ($224 \times 224$)**. Our conclusions on "adaptation bias" are drawn based on this rigorous process, with detailed analysis as follows:
>
> - **Intrinsic Bias, Not a Preprocessing Issue**: The "adaptation bias" we describe is not a resolution mismatch caused by feeding $32\times32$ tensors into a $224 \times 224$ model, but a fundamental distribution shift. Upscaling a  $32\times32$  image to $224 \times 224$ cannot recover the high-frequency information and semantic clarity inherent in the original pre-training data (internet-scale, high-resolution images).
> - **Empirical Evidence**: This claim is validated in Figure 3 and Section 4.2. When evaluating the model on STL-10 at varying resolutions (by downsampling high-resolution images), we observed that "as resolution increases, model performance consistently improves and adaptation bias gradually diminishes." This confirms that the performance drop stems from differences in the information content of the domain (resolution shift) rather than implementation flaws.
>
> Regarding the performance discrepancy compared to the original CLIP paper: **We conducted tests using the official CLIP model (downloaded from [1]) and the ZSCLIP code from the PromptSRC repository**. The consistent performance improvements across CLIP, CoOp, and PromptSRC demonstrate the reliability of our reproduction. For the observed partial performance difference, we found in the official CLIP GitHub repository [2] that OpenAI did not release the official testing code. Based on the provided test prompts [3] and Jupyter Notebook files, CLIP computes the average of text features from multiple prompt templates—unlike our approach of using a single prompt template ("a photo of a _"). **This discrepancy in testing protocols is the likely cause of the performance difference.** In summary, our reproduction is methodologically sound, and the observed performance gap with CLIP arises from differences in testing procedures.
>
> [1] https://openaipublic.azureedge.net/clip/models/5806e77cd80f8b59890b7e101eabd078d9fb84e6937f9e85e4ecb61988df416f/ViT-B-16.pt
>
> [2] https://github.com/openai/CLIP
>
> [3] https://github.com/openai/CLIP/blob/main/data/prompts.md
>
> **W5: Response to "Incorporate MLLM-related Settings into the Main Text"**
>
> A: We are pleased that your concerns regarding W5 have been addressed. We agree with your viewpoint and will supplement the relevant experimental settings, including the MLLM’s model architecture, pre-training version, fine-tuning parameters, and detailed configurations of comparative experiments, into the "Experimental Design" section of the main text.

---

### Author Response · Authors · 2025-12-01
**Final Remarks by Authors**

Dear SAC, AC, and Reviewers,

We sincerely thank you for the constructive feedback and productive discussions. Through the rebuttal process, we have addressed concerns regarding medical imaging performance and the discussion on weakly supervised learning. Below, we summarize the significance of this work as explicitly highlighted by the reviewers:

1. Timeliness and Strategic Value: Reviewer bHUs identifies this work as "an important position paper" for the field, and Reviewer c4Ht acknowledges it is a "timely study in the era of large models." It addresses a critical gap by systematically comparing traditional SSL with modern pre-training paradigms.

2. Comprehensive and Convincing Evaluation: Both Reviewers bHUs and Gqrj commend the experiments as "comprehensive" across multiple settings (standard, open-set, open-world, long-tailed), with Reviewer bHUs further concluding that the results are "convincing."

3. Actionable Insights for the Community: Reviewer c4Ht notes that our study has the "potential to deliver useful insight for the SSL community." By delineating boundary conditions, we provide practical guidance that Reviewer bHUs describes as "valuable conclusions."

Unfortunately, we have not received further responses from the following reviewers:

- Reviewer Gqrj: The negative assessment primarily stems from the issue of architectural alignment. We have clarified both in the paper and in our response that different methods possess specific "optimal" architectures, and larger models do not necessarily yield better performance in all contexts. Notably, the reference [1] cited by the reviewer similarly does not strictly align architectures, validating that our experimental setting is reasonable.
- Reviewer c4Ht: Following their suggestion, we conducted comparative experiments on medical datasets. The results fully align with our conclusions regarding the "adaptation bias" of pre-trained models. Consequently, we believe we have effectively resolved their concerns.
In addition, we have revised the manuscript according to the results of the Rebuttal, incorporating the constructive suggestions from the reviewers. Please refer to the table of revisions at the end of this letter.
Conclusion: In summary, our work systematically evaluates the trade-offs between pre-trained models and SSL under fixed label budgets, aiming to guide researchers in the foundation model era and advocating for hybrid approaches. We respectfully request the Area Chair to comprehensively consider the overall feedback from the reviewers and the practical application scenarios when assessing our work's contribution to the community.

We sincerely appreciate your time and efforts throughout the review process.

Best regards,

The Authors of Submission 7026

Reference: [1] Delving into Weakly Supervised Learning with Pre-Trained Models, https://openreview.net/forum?id=RgWATMmWmz

Sec. 3: Enriched the existing experimental settings and added experimental setups for comparison between VLM (Vision-Language Model) and MLLM (Multimodal Large Language Model); added a module/section to discuss the rationality of the default architecture.

Sec. 4.2: Incorporated medical data into the standard SSL experiments and analyzed the impact of adaptation bias caused by domain shift/change.

Conclusion: Proposed to explore a comparison between weak supervision and pre-trained models as a direction for future work.

---

### Meta-Review · Area_Chair_2vdU · 2026-01-09

**Summary:**

The paper received divergent scores (2, 4, 8). Upon reviewing the content and feedback, the AC acknowledges the study addresses practically significant questions relevant to real-world applications. However, the paper suffers from unfair comparisons, and from a practical standpoint, combining pretrained models with SSL represents a critical solution avenue that warrants deeper exploration. Weighing all reviewer opinions, the AC recommends rejection, suggesting the authors strengthen the comparative experiments and resubmit to future conferences.

**Reviewer Concerns:**

Addressed Concerns:
- The paper should report MLLM performance with no-thinking or direct-answering prompts to better reflect their true classification capabilities.
- Exploratory experiments on low-overlap scenarios (e.g., medical scenarios) are lacking.
- Some studies on semi-supervised learning with CLIP are not discussed.
- The experiments overlook that SOTA SSL methods like FixMatch outperform pre-trained models on CIFAR-10/100, making the conclusion about pre-trained models’ superiority incomplete.
- Different model architectures lead to unfair experimental settings

Outstanding Concerns:
- Unsound comparison and meaningless conclusions.
- SSL with CLIP has already been explored, and under such unfair experimental conditions, the conclusions drawn from the experiments are not meaningful.

**Reviewer Scores:**

The paper received divergent scores (2, 4, 8). The concerns from the 4- and 8-score reviewers were adequately addressed, but the 2-score reviewer maintained that most of their concerns remained unresolved.

---

### Decision · Program_Chairs · 2026-01-26

Reject